# The Cell-Specific Effects of JAK1 Inhibitors in Ulcerative Colitis

**DOI:** 10.3390/jcm14020608

**Published:** 2025-01-18

**Authors:** Suzanne H. C. Veltkamp, Philip W. Voorneveld

**Affiliations:** Department of Gastroenterology and Hepatology, Leiden University Medical Center, 2333 ZA Leiden, The Netherlands; s.h.c.veltkamp@lumc.nl

**Keywords:** ulcerative colitis, inflammatory bowel disease, JAK inhibitors, colon, cell-specific effects

## Abstract

JAK1 inhibitors have become an important addition to the therapeutic options for ulcerative colitis (UC), targeting key inflammatory pathways mediated by cytokines such as the IL-6 family, interferons, IL-2 family, IL-10 family, and G-CSF. However, not all patients respond equally, and chronic inflammation persists in a subset of individuals. The variability in treatment response may reflect the heterogeneity of UC. Immune cells, epithelial cells, and stromal cells may have distinct contributions to disease pathogenesis. While JAK inhibitors were originally designed to target immune cells, their impact on non-immune cell types, such as epithelial and stromal cells, remains poorly understood. Investigating the mechanisms through which JAK1 inhibitors affect these diverse cellular populations and identifying the factors underlying differential responses is crucial to optimizing outcomes. This review explores the roles of immune, epithelial, and stromal cells in response to JAK1 inhibition and discusses potential strategies to improve treatment precision, such as predicting responders and identifying complementary therapeutic targets.

## 1. Introduction

Ulcerative colitis (UC) is a chronic inflammatory bowel disease (IBD) affecting the mucosa and submucosa of the colon. In 2023, the estimated global prevalence of UC was 5 million cases, with incidence rates continuing to rise [1]. The primary goal of UC treatment is to induce and sustain remission, a process that is often prolonged, requiring continuous therapy that can be both costly and complex [2,3]. Current treatments for UC include non-specific immune suppressants such as aminosalicylates (5-ASAs), azathioprine, and methotrexate, as well as biologics that target specific molecules like TNF-α, IL-12/IL-23, or integrins. While these therapies are effective for many patients, a significant number either do not respond to treatment or lose response over time, leading to persistent chronic inflammation.

A relatively new treatment option for UC is Janus kinase (JAK) inhibitors, which are orally administered small-molecule therapies. The JAK family, consisting of JAK1, JAK2, JAK3, and TYK2, plays a pivotal role in mediating the signaling pathways of over 50 cytokines [4]. Cytokine binding to its receptor induces receptor dimerization and activation of receptor-associated JAKs through transphosphorylation. These activated JAKs then phosphorylate the receptor, creating a docking site for members of the signal transducer and activator of transcription (STAT) family. JAKs subsequently phosphorylate STATs, which form dimers and translocate to the nucleus to affect gene transcription. 

While some JAK inhibitors, such as tofacitinib, target multiple JAK family members (predominantly JAK1 and JAK3, and to a lesser extent JAK2 and TYK2), others, including filgotinib and upadacitinib, are more selective for JAK1. These inhibitors act on the ATP binding site, which is highly conserved across the JAK family, so higher doses can lead to inhibition of additional JAK members [5]. Increased selectivity might contribute to a better safety profile. For instance, tofacitinib has been associated with a higher risk of herpes zoster infection compared to filgotinib in patients with immune-mediated inflammatory diseases [6]. Even selective JAK1 inhibitors, however, can impact multiple inflammatory pathways, as JAK1 mediates signaling for IL-6 family cytokines, interferons, IL-2 family cytokines, IL-10 family cytokines, and G-CSF [7]. Despite the broad impact of JAK inhibitors, a significant proportion of patients fails to respond to this drug as well.

The failure to respond to JAK inhibitors may be attributed to disease heterogeneity [8]. Multiple cell types play a role in the pathogenesis of UC, including immune cells, epithelial cells, and stromal cells, all of which may play a role in the observed variation in disease outcomes [9]. While JAK1 inhibitors were initially designed to modulate immune cell activity, they potentially also affect other cell types involved in UC pathogenesis [10,11,12]. The precise mechanisms by which JAK1 inhibitors influence the diverse cell types in UC patients and the reasons for the differential responses to treatment remain unclear. This review aims to address these gaps in knowledge by exploring the current understanding of the effects of JAK1 inhibitors on immune, epithelial, and stromal cells in UC. It further discusses how these cell-specific effects may explain the variability in patient responses to JAK inhibitors. Understanding these mechanisms may enable the prediction of patient response to JAK inhibitors and identify complementary targets to enhance efficacy in non-responders.

## 2. The Effect of JAK1 Inhibitors on Immune Cells

In the pathogenesis of ulcerative colitis (UC), diverse myeloid and lymphoid cells produce and respond to a wide array of cytokines [13]. Clinical studies analyzing inflamed colon samples of UC patients have found increased mRNA expression of IFN-γ, OSM, TNF-α, IL-6, IL-10, IL-12, IL-13, IL-17, IL-21, IL-22, IL-23, and IL-33 compared to healthy subjects [14,15,16]. On a protein level, IFN-γ, TNF-α, IL-1β, IL-5, IL-8, IL-13, and IL-17 are upregulated in UC patients [17]. While OSM protein expression is also elevated, the increase is not statistically significant [17]. Among these cytokines, IFN-γ, IL-6, IL-10, IL-21, IL-22, and OSM signal through JAK1, driving diverse processes such as neutrophil recruitment, promotion of neutrophil and T cell (including regulatory T cell (Treg)) survival, stimulation of antigen presentation by dendritic cells, activation and differentiation of T cells and macrophages, and stimulation—but also inhibition—of B cell proliferation [18,19,20,21]. Theoretically, JAK1 inhibition should suppress these processes, although the overlapping and sometimes opposing effects of these cytokines complicate predictions of therapeutic outcomes.

Filgotinib, a preferential JAK1 inhibitor, has been shown to inhibit differentiation of naïve CD4 T cells from healthy subjects to Th1, Th2, and Th17 cells in vitro [22]. Both filgotinib and upadacitinib reduce proliferation, increase apoptosis, and decrease CD25 (an activation marker) expression in peripheral blood mononuclear cells (PBMCs) from healthy subjects [23]. Furthermore, upadacitinib inhibits GM-CSF-induced IL-1β production by THP-1 cells (a human monocyte cell line) and human neutrophils. It also inhibits production of caspase-1(p20), which is released by human neutrophils during NLRP3 inflammasome activation [24]. Another JAK1 inhibitor, GLPG0555, suppresses CXCL10, but increases IL-6 secretion in human monocytes that have been polarized towards an M1 phenotype in vitro. This JAK1 inhibitor also increases the CD163/CD64 ratio in these cells, suggesting a shift toward an M2-like phenotype [25].

The effects of JAK inhibition on specific immune cell types have been studied in vivo. In patients with Crohn’s disease (CD) treated with upadacitinib, Aguilar et al. observed significant downregulation of genes associated with inflammatory monocytes, CD8+ T cells, CD4+ T cells, dendritic cells, plasma cells, B cells, macrophages, and mast cells in colon biopsies of responders who achieved endoscopic remission [26]. Key T cell markers (IFNG, GZMH, and TBX21), plasma cell marker DERL3, and inflammatory monocyte markers (OSM and S100A8) were also significantly reduced, alongside mast cell markers ADCYAP1, HDC, and TPSAB1. In contrast, these markers remained unchanged in non-responders. Interestingly, these T cell, plasma cell, and mast cell markers were not downregulated in colon biopsies of CD patients who were treated with anti-TNF and reached endoscopic remission, suggesting a treatment-specific effect rather than remission alone. Differences in treatment-induced gene expression changes between upadacitinib and anti-TNF can partially be attributed to baseline expression level differences. In the upadacitinib group, prior anti-TNF exposure in 98% of patients influenced baseline expression patterns. The effects of JAK inhibitors on immune cell composition specifically in patients previously exposed to other biologics such as anti-α4β7-integrin and anti-IL12/23 remain unclear. A notable limitation of the study by Aguilar et al. is its reliance on bulk RNA sequencing of intestinal biopsies, which does not provide direct insights into individual cell types. To address this, the authors integrated the bulk RNA-seq data with publicly available single-cell RNA sequencing (scRNA-seq) datasets, using the latter to infer cell population contributions. However, it should be noted that findings in CD patients might not be directly applicable to UC patients. Although both CD and UC fall under the umbrella of inflammatory bowel diseases (IBDs), their etiologies differ, and distinct mechanisms contribute to their pathogenesis [27].

In UC patients, van Gennep et al. analyzed the effect of tofacitinib (a pan-JAK inhibitor) on immune cell abundance in colon biopsies and found significant reductions in all immune cell populations in responders, including CD4 and CD8 T cells, Tregs, neutrophils, and macrophages [28]. In non-responders, only macrophages and neutrophils were significantly reduced. A separate scRNA-seq study by Melón-Ardanaz et al., not yet peer-reviewed at the time of writing, revealed that tofacitinib reduced JAK/STAT pathway activity across most immune cell types in responders [29]. This was accompanied by a decrease in the abundance of Tregs, naïve T cells, B cells, plasma cells, neutrophils, eosinophils, inflammatory monocytes, dendritic cells, and INHBA+ macrophages, but not CD4 and CD8 T cells, and mast cells were slightly upregulated. These findings align with those of van Gennep et al. and Aguilar et al., with some exceptions, such as the effects on CD4 and CD8 T cells and mast cells. In non-responders, Melón-Ardanaz et al. observed an increase in naïve T cells, B cells, and myeloid cells, including neutrophils, inflammatory monocytes, dendritic cells, and INHBA+ macrophages. These changes in cell abundance in non-responders were not observed by van Gennep et al. or Aguilar et al. [26,28]. A possible explanation for this discrepancy in results is the use of different JAK inhibitors, differences in patient population (UC and CD), and differences in the definition of response.

Melón-Ardanaz et al. further investigated the effects of tofacitinib on macrophages [29]. Although these cells did not exhibit the largest changes in JAK/STAT pathway activity and abundance, they did show distinct alterations in NF-κB signaling depending on the treatment response. In responders, NF-κB pathway activity decreased following tofacitinib treatment, whereas it increased in non-responders, indicating opposing effects on macrophage activation states. Among macrophage subsets, FOLR2+ macrophages showed the most divergent gene expression patterns, adopting an M2-like phenotype in responders and an M1-like phenotype in non-responders. In vitro experiments demonstrated that tofacitinib enhanced macrophage activation in response to LPS, potentially through disruption of autocrine anti-inflammatory IL-10 signaling. Non-responders exhibited higher baseline IL-10 pathway activity, suggesting that interference with this signaling by tofacitinib may contribute to persistent inflammation in these patients. These results illustrate that not all cell states respond equally to JAK inhibition, which highlights the complexity of the effects of JAK inhibitor treatment. 

Two randomized controlled trials explored the effects of specific JAK1 inhibition in UC, reporting no significant changes in blood neutrophil and lymphocyte counts upon filgotinib treatment, whereas upadacitinib was associated with an increased risk of neutropenia [30,31]. The specific changes in colonic immune cell populations upon JAK1 inhibitor treatment in UC are yet to be elucidated.

## 3. The Effect of JAK1 Inhibitors on Colon Epithelial Cells

Epithelial cells play a role in the pathogenesis of UC by maintaining intestinal barrier integrity, regenerating damaged tissue, and regulating immune responses. The effects of JAK inhibitors on these processes have been studied in both in vitro and in vivo models. In addition, several studies have examined the effects of JAK inhibition on colon epithelial cell abundance and JAK/STAT signaling in IBD patients. Notably, JAK inhibitors are now also being evaluated as a therapeutic option for celiac disease, another condition where intestinal epithelial cells play a key role in pathogenesis [32].

### 3.1. The Effect of JAK Inhibitors on Epithelial Cells in IBD Patients

Aguilar et al. investigated the cell-specific effects of upadacitinib in CD patients using whole-biopsy RNA sequencing [26]. The most prominent changes were in genes associated with enterocytes. In addition, genes associated with goblet cells, enteroendocrine cells, tuft cells, and transit-amplifying (TA) cells were upregulated. Several genes linked to enterocytes and the secretory and stem/TA compartment were significantly upregulated in patients who achieved endoscopic remission. Colonocyte marker AQP8 and goblet cell marker RTNLB were upregulated in responders to both upadacitinib and anti-TNF treatment, suggesting that these changes may not be treatment-specific.

Melón-Ardanaz et al. explored the effects of tofacitinib, a pan-JAK inhibitor, on UC patients using single-cell RNA sequencing [29]. They observed a reduction in JAK/STAT pathway activity in epithelial cells of responder patients after treatment. In addition, the abundance of all epithelial cell types (except APOA4+ cells) increased in responders, but not in non-responders. Other studies found that the number of epithelial cells staining positive for STAT3 and pSTAT3 was decreased after tofacitinib treatment [28,33]. Interestingly, JAK1, JAK2, and STAT3 epithelial expression levels did not correlate with the degree of inflammation [28].

These findings demonstrate that JAK inhibitors induce changes in mRNA and protein expression and increase epithelial cell abundance in the colons of responder patients. However, these studies did not investigate functional changes of epithelial cells upon JAK inhibitor treatment.

### 3.2. The Effect of JAK1 Inhibitors on Barrier Integrity

Intestinal barrier integrity is a critical factor in UC pathogenesis, and many IBD-associated single-nucleotide polymorphisms (SNPs) are linked to barrier function regulation [34]. Impaired barrier integrity permits paracellular translocation of luminal microbial antigens into the lamina propria, activating immune responses and perpetuating inflammation, ultimately leading to mucosal damage. For example, the pANCA autoantibody, commonly associated with UC, exhibits cross-reactivity to the bacterial outer membrane protein OmpC [35]. pANCA autoantibodies are detectable in the serum of some individuals prior to a formal diagnosis of UC; however, this antibody lacks sufficient predictive performance to serve as a biomarker of UC onset [36,37]. Nevertheless, their presence prior to diagnosis, similar to ASCA antibodies in undiagnosed CD and celiac disease, suggests that increased intestinal permeability may contribute to the early stages of pathogenesis [37,38,39].

In UC patients, claudin 2 expression is often increased, increasing permeability, while other tight junction proteins, such as claudin 3, claudin 4, zonula occludens 1 (ZO-1), occludin, tricellulin, and desmoglein 2, which decrease permeability, are downregulated [40,41,42,43,44]. Furthermore, treatment of mice with dextran sodium sulfate (DSS), a chemical agent that induces colitis, downregulates expression of occludin, a tight junction protein, in intestinal epithelial cells (IECs) in a phospholipase D2 (PLD2)-mediated pathway [45]. Inhibition of this pathway can rescue mice from DSS-induced colitis, implicating the integrity of epithelial tight junctions in the pathogenesis of UC. Many cytokines signaling through JAK1 affect intestinal barrier integrity [46]. Some of these cytokines, including IFN-γ, IL-4, and IL-13, decrease barrier integrity, while others, including IL-10 and IL-15, increase barrier integrity, and others, including IL-6, IL-22, and IL-23, have ambiguous effects on barrier integrity.

Currently, there is no direct evidence regarding how JAK1 inhibition affects intestinal barrier integrity. However, several studies have explored the effects of pan-JAK inhibitors such as tofacitinib on barrier integrity. These studies commonly assess paracellular flux of macromolecules (e.g., 4 kDa FITC-dextran (FD4) or 0.4 kDa Lucifer yellow) and transepithelial electrical resistance (TEER) as proxies for barrier integrity. Tofacitinib and other pan-JAK inhibitors have been shown to block permeability increases induced by IFN-γ, IL-22, and other inflammatory cytokines in various models, including T84 cells, Caco-2 cells, and organoids derived from healthy subjects or UC patients [47,48,49]. In Caco-2 cells co-cultured with PTPN2-deficient macrophages, tofacitinib blocks the reduction in TEER, but not the increase in permeability to FD4 [50]. PTPN2 is a tyrosine phosphatase that negatively regulates the JAK/STAT pathway.

In all cell line models, these changes in barrier integrity are positively correlated with changes in claudin 2 expression [47,48,49,50]. Claudin 2 is transcriptionally regulated by STAT1, which binds to its promoter. Tofacitinib reduces claudin 2 promoter activity by inhibiting the JAK/STAT pathway, thereby improving barrier integrity [47]. The expression of other tight junction proteins, including ZO-1; occludin; tricellulin; and claudins 1, 4, and 15, is not affected by IFN-γ or tofacitinib treatment in T84 cells. Instead, tofacitinib reduces the number of intercellular gaps caused by IFN-γ-induced ZO-1 relocalization [47]. Caco-2 cells co-cultured with PTPN2-deficient macrophages express less claudin 4, JAM-A, occludin, and tricellulin [50]. Tofacitinib normalizes the expression of occludin and tricellulin, but not claudin 4 and JAM-A. Interestingly, in a mouse model lacking Ptpn2 in monocytes and macrophages, protein expression of claudin 4, JAM-A, and occludin was reduced, but restored by tofacitinib, while tricellulin remained unaffected [50]. This shows that the effects of cytokines and JAK inhibitors vary across different models, underscoring the need for studies using materials from patients who have been treated with JAK inhibitors to fully understand their impact in UC.

In addition to tight junction regulation, epithelial barrier integrity can also be altered via loss of cells by shedding or cell death. Filgotinib and tofacitinib restore viability of murine small intestinal organoids treated with IFN-λ. Tofacitinib also restores barrier function and decreases epithelial cell death in Casp8^ΔIEC^ mice overexpressing IFNL [51]. In colorectal cancer cell lines such as HT-29 cells, IFN-γ + TNF-α-induced cell death is driven by the JAK1/2-STAT1 pathway and supported by the non-enzymatic scaffold function of caspase-8. Upadacitinib can, in contrast to filgotinib, prevent IFN-γ + TNF-α-induced killing in HT-29, DLD-1, HCT-15, SW-48, and SW-948 cells and CD patient-derived colon organoids [52].

In conclusion, while studies have demonstrated that pan-JAK inhibitors like tofacitinib can improve intestinal barrier integrity in various models, the specific effects of JAK1 inhibitors on these processes remain to be elucidated.

### 3.3. The Effect of JAK1 Inhibitors on Regeneration and Wound Healing

Epithelial regeneration and wound healing are critical processes in achieving mucosal healing for patients with ulcerative colitis (UC). Successful mucosal healing is strongly associated with reduced colectomy rates and a lower risk of clinical relapse, emphasizing its importance in long-term disease management [53,54]. While the specific effects of JAK1 preferential inhibitors on these processes remain unexplored, insights can be gained from studies investigating pan-JAK inhibitors like tofacitinib.

Tofacitinib has been shown to impair wound healing in certain experimental settings. For example, it inhibits wound closure in a scratch assay with Caco-2 cells and exacerbates colonic damage in DSS-treated mice when administered during the early repair phase [10,55]. Similarly, high doses of tofacitinib (8 μM in 2% carboxymethyl cellulose (CMC), given orally once daily) significantly impair the healing of colon wounds created with biopsy forceps, while lower doses (2 μM in 2% CMC) have no significant impact [55]. These findings suggest a dose-dependent influence of JAK inhibition on wound repair processes.

A key pathway involved in epithelial repair is the JAK/STAT signaling cascade, particularly through STAT3 activation. In murine models, STAT3 deficiency (Stat3^ΔIEC^) leads to delayed wound healing, increased tissue damage, reduced epithelial proliferation, and heightened apoptosis in DSS-induced colitis [56,57]. Additionally, the expression of genes associated with Lgr5+ intestinal stem cells (ISCs), which are crucial for epithelial regeneration, is significantly diminished in these mice [58]. Given that JAK1 is a primary upstream activator of STAT3, inhibiting JAK1 could potentially mimic the effects of STAT3 knockout.

Cytokines such as IL-22 and IL-10 promote epithelial proliferation and regeneration through STAT3 activation. IL-22, produced by CD11c+ cells in a murine colitis model, stimulates organoid growth from wild-type mouse intestinal crypts, while Stat3-deficient crypts fail to generate organoids [56,57,58]. IL-22 promotes similar effects in human colon organoids, including transcriptional changes like downregulation of the differentiation marker KRT20 and upregulation of inflammation-associated genes. These effects are blocked by tofacitinib, indicating that JAK inhibition can interfere with cytokine-mediated regeneration [10]. IL-10 increases clonogenicity and the number of Lgr5+ ISCs in mouse small intestinal organoids, while IL-13 reduces this [59].

Interestingly, the role of IFN-γ in epithelial regeneration is less clear, with studies reporting contradictory findings. While one study found no effect of IFN-γ on organoid regeneration, another demonstrated that IFN-γ induces reserve intestinal stem cells characterized by telomerase expression, suggesting a positive role in regeneration [11,59]. Tofacitinib blocks this induction, highlighting its potential to modulate the impact of IFN-γ on epithelial repair. In contrast, another study showed that IFN-γ decreases the number of Lgr5+ ISCs, induces pro-apoptotic transcriptional changes, and reduces the outgrowth of human and mouse small intestinal organoids [60]. These detrimental effects are blocked by ruxolitinib, a JAK1/2 inhibitor, and are absent in JAK1-knockout organoids, further implicating JAK1 in mediating these responses.

Most evidence regarding JAK inhibitors’ effects on wound healing comes from murine models. However, significant differences in human and murine intestinal biology underscore the need for studies using patient-derived materials. For example, human colon stem cells exhibit resilience after injury, persisting and resuming proliferation without reliance on de-differentiation. In contrast, mouse ISCs are eradicated following radiation injury and are replaced by de-differentiated neighboring cells [61]. In conclusion, current evidence suggests that JAK/STAT pathway inhibition has predominantly negative effects on epithelial regeneration and wound healing. However, further research is needed to determine the impact of JAK1 inhibitors specifically.

### 3.4. The Effect of JAK1 Inhibitors on Inflammatory Functions of Epithelial Cells

In addition to their roles in barrier integrity, regeneration, and wound healing, intestinal epithelial cells actively participate in the inflammatory process. These cells produce a variety of chemokines and cytokines to recruit and activate immune cells, thereby amplifying inflammatory responses.

Epithelial cells from UC patients and healthy subjects express increased levels of TNF-α and IL-8 mRNA upon TNF-α stimulation, which can be suppressed by tofacitinib [62]. Both TNF-α and IL-8 are central to neutrophil recruitment and the amplification of inflammation in UC. These findings highlight the role of JAK inhibitors in mitigating epithelial-driven inflammatory cascades. Colonic mucosal explants from UC patients, which are ex vivo cultured primary tissue samples, express NOS2. This gene encodes inducible nitric oxide synthase [iNOS], an enzyme responsible for producing nitric oxide that is upregulated in the inflamed intestinal epithelium of IBD patients [63,64]. Tofacitinib inhibits NOS2 expression in these mucosal explants. Tofacitinib, as well as upadacitinib (JAK1 inhibitor) and brepocitinib (TYK2/JAK1 inhibitor), also inhibits IFN-γ-induced pSTAT1 and iNOS expression levels in colon organoids from healthy subjects.

Epithelial responses to viral and cytokine stimuli further illustrate the inflammatory functions of IECs. In HT-29 cells, viral mimic poly(I:C) and TNF-α induce IRF9, ISG15, and IFN-β, key proteins in antiviral and inflammatory pathways. These responses, driven by the JAK1-pSTAT-IRF9 pathway, are inhibited by JAK1-specific inhibitors such as filgotinib [65]. Additionally, during active UC, epithelial cells exhibit elevated MHC-II expression, which promotes T cell activation [59,66]. Tofacitinib suppresses the TNF-α and poly(I:C)-induced upregulation of MHC-II in UC patient-derived organoids, potentially reducing epithelial-driven T cell responses [67].

In conclusion, JAK inhibitors suppress key inflammatory functions of IECs, including cytokine production, NO synthesis, antiviral responses, and antigen presentation. Evidence from studies on pan-JAK inhibitors like tofacitinib, as well as preferential JAK1 inhibitors such as upadacitinib and filgotinib, underscores the potential of JAK1-targeted therapies to mitigate epithelial inflammation in UC.

### 3.5. The Effect of JAK1 Inhibitors on Colitis-Associated Colorectal Cancer

UC patients have an increased risk of developing colorectal cancer, as indicated by a meta-analysis reporting an odds ratio of 1.51 (95% confidence interval (CI) 1.28–1.74) [68]. Chronic inflammation drives neoplastic processes, including cell proliferation, survival, and migration. STAT3 activation in intestinal epithelial cells by IL-6, IL-11, IL-21, and IL-22 enhances colon cancer development in mice [69,70,71]. By inhibiting cytokine signaling and reducing inflammation, JAK inhibitors may decrease the risk of colitis-associated colorectal cancer. However, this suppression of the immune response may simultaneously impair immune surveillance, potentially increasing cancer risk. A meta-analysis reported a relative risk of 1.05 (95% CI 0.47–2.35) for nonmelanoma skin cancer and 1.39 (95% CI 0.68–2.85) for other malignancies in patients with inflammatory diseases who were treated with JAK inhibitors [72]. Clinical trials specifically investigating JAK1 inhibitors in UC patients have not demonstrated an elevated risk of malignancies, although their statistical power and follow-up time may be insufficient to reliably detect such rare events [30,31]. Additionally, hyperactivation of the JAK/STAT pathway has been implicated in drug resistance of tumor cells, but JAK1 inhibitors can enhance the potency of therapies against non-small cell lung cancer [73,74,75]. Further research is needed to clarify the effects of JAK1 inhibition specifically on colitis-associated colorectal cancer.

## 4. The Effect of JAK1 Inhibitors on Colonic Stromal Cells

Stromal cells are increasingly recognized as key players in the pathogenesis of UC, although our understanding of their exact roles remains limited. Recent advances, such as single-cell RNA sequencing (scRNA-seq), have begun to shed light on the heterogeneity of colonic stromal cells and their involvement in UC. For instance, Kinchen et al. identified several stromal cell clusters in UC patients, including clusters S1–S4 and myofibroblasts [12]. Among these, the S4 cluster was significantly expanded in UC patients compared to healthy controls. This cluster is enriched for gene ontology (GO) terms such as “response to tumor necrosis factor”, “positive regulation of leukocyte migration”, and “response to bacterium”. It displays upregulation of pro-inflammatory genes such as TNFSF14, CCL19, CCL20, CD74, and IL-33 [12]. Similarly, Gavriilidis et al. demonstrated that primary intestinal fibroblasts (PIFs) from UC patients express elevated IL-8 mRNA and protein levels compared to healthy controls [76]. As a chemoattractant, IL-8 recruits neutrophils and induces neutrophil extracellular trap (NET) formation, further linking stromal cells to immune cell recruitment and inflammation [77].

Studies by Melón-Ardanaz et al. and Aguilar et al. have highlighted the nuanced and context-dependent effects of JAK inhibitors on stromal cells. In UC patients treated with tofacitinib, Melón-Ardanaz et al. observed an overall increase in stromal populations in responders, with all fibroblast subsets except inflammatory fibroblasts showing significant expansion [29]. S1 fibroblasts, characterized by high expression of ADAMDEC, exhibited the largest number of differentially expressed genes, many of which were involved in immune modulation and tissue remodeling. In contrast, Aguilar et al. found that, in colon biopsies of CD patients treated with upadacitinib who reached endoscopic remission, many genes linked to crypt fibroblasts were significantly downregulated [26]. For example, THY1 (a crypt fibroblast marker) and COL3A1 (a pan-fibroblast marker) were downregulated, while crypt fibroblast markers SOX6, PTGDR2, and PDGFD were unaffected in responders. Upadacitinib also significantly downregulated CHI3L1, an inflammatory fibroblast marker, in responders. This finding is in line with what Melón-Ardanaz found in UC patients treated with tofacitinib.

Inflammatory fibroblasts in IBD are known to highly express OSMR, a receptor that signals via JAK1, making this cell population, in theory, particularly sensitive to JAK inhibition [78]. Subepithelial myofibroblasts from UC patients also express increased OSMR mRNA levels compared to healthy subjects [79]. In vitro studies show that stimulation of these fibroblasts with OSM after IL-1a and TNF-α prestimulation (which upregulates OSMR expression) leads to higher expression of the chemotactic factors CCL2, CXCL9, CXCL10, and CXCL11 [79]. These factors play key roles in immune cell recruitment and inflammation. In addition to OSMR, colon fibroblasts also express other JAK1-dependent receptors, such as high levels of *IL11ra* and *IL6st* (encoding gp130) [80]. Colon fibroblasts also express IL-11 [78,80]. Lim et al. showed that fibroblast-selective expression of *IL11* in a genetic mouse model causes colon shortening and elevates fecal calprotectin, which are features of experimental colitis [81]. However, Nishina et al. found that deficiency of *IL11ra1* and *IL11* exacerbates DSS-induced colitis [82]. In addition, IL-11 protein levels are increased in the colons of patients with mild UC, but decreased in patients with severe UC [83]. These conflicting results suggest that the role of IL-11 in UC may be altered depending on the tissue microenvironment and the duration of its production. This variability complicates predictions regarding the outcomes of targeting IL-11 signaling through JAK1 inhibition.

The contrasting observations regarding total fibroblast abundance in IBD patients treated with JAK inhibitors alongside the diverse effects of cytokines illustrate the complexity of stromal cell contributions to UC pathogenesis. These findings highlight the nuanced effects of JAK1 inhibition on stromal cells. Despite these complexities, there is a consensus that inflammatory fibroblasts, which prominently express receptors for OSM and IL-11—both signaling via JAK1—are particularly sensitive to JAK inhibition. While evidence suggests that JAK inhibitors reduce the abundance of inflammatory fibroblasts, their effects on other stromal cell subsets remain unclear. These findings underscore the need for further investigation to fully elucidate the role of stromal cells and JAK1 inhibitors in UC.

## 5. The Roles of Various Cell Types in Resistance to JAK1 Inhibitor Treatment in UC

Not all IBD patients respond to treatment with JAK inhibitors, and the biological mechanisms behind this are unknown. Therefore, it is currently difficult to predict which patients will respond to therapy. This unpredictability increases medical costs and exacerbates the disease burden for patients. However, recent studies have identified some baseline differences between responders and non-responders to JAK inhibitor therapy, offering potential insights into the mechanisms of resistance. Melón-Ardanaz et al. found that, in UC patients treated with tofacitinib, baseline JAK/STAT pathway activity was significantly higher in T cells, B cells, mast cells, fibroblasts, and endothelial cells in responders compared to non-responders [29]. Interestingly, no significant baseline differences in JAK/STAT activity were observed in the colon epithelium. This finding was corroborated by other studies. Baseline protein levels of JAK1, JAK2, and STAT3, which were mostly expressed in the epithelium, along with epithelial pSTAT3 levels, were not significantly associated with tofacitinib response in UC patients [28,33]. Similarly, a post hoc analysis of the FITZROY trial found that epithelial pSTAT1 and pSTAT3 levels did not predict the endoscopic or histologic response to filgotinib, a JAK1 inhibitor, in CD patients [84]. Aguilar et al. also reported that no baseline gene signature could reliably predict the response to upadacitinib in CD patients [26]. These findings suggest that baseline JAK/STAT signaling in immune and stromal cells, rather than epithelial cells, may determine the treatment response.

Resistance mechanisms may also involve alternative pathways. For example, autocrine IL-10 signaling via TYK2 and JAK1 in macrophages, as discussed earlier, may contribute to persistent inflammation in non-responders. Furthermore, non-response to tofacitinib in UC patients has been associated with increased baseline NF-κB pathway activity in colon mucosa cells [29].

The expression of drug transporters in specific cell types may also influence resistance to JAK inhibitors. For example, tofacitinib is transported by multidrug and toxin extrusion protein 1 (MATE1), and MATE1 expression in the colon has been linked to treatment response in UC patients [85,86]. In patient-derived organoids, MATE1 expression reflects its expression in colonic sections, and the ability of tofacitinib to rescue organoid viability after cytokine treatment correlates with patient treatment outcomes [86]. These findings suggest that epithelial cells and their transporter expression may play a role in determining treatment efficacy, although this requires further validation in larger cohorts.

Whether transporter expression influences resistance to JAK1 inhibitors remains to be elucidated. Filgotinib is a substrate of the P-glycoprotein (P-gp) transporter, but its exposure is not impacted by P-gp inhibitors or inducers in healthy subjects [87]. Upadacitinib is a substrate for P-gp and breast cancer resistance protein (BCRP), but its high permeability and solubility reduce the impact of transporter modulation on drug exposure [88]. Nevertheless, transporter expression in specific cell types could still affect the efficacy of these JAK1 inhibitors, although this possibility remains unexplored. Elevated P-gp expression in peripheral blood lymphocytes has been associated with general therapy resistance in UC [89]. In addition, P-gp-expressing Th17 cells are enriched in the intestines of CD patients, and in vitro, these cells are resistant to glucocorticoid treatment [90]. However, P-pg inhibition does not sensitize the cells to glucocorticoid treatment, suggesting that the function of P-gp is not central to their steroid-resistant phenotype.

Resistance to JAK inhibitor therapy in UC likely involves factors beyond JAK/STAT signaling and transporter expression. One important consideration is the activation of alternative inflammatory pathways, such as TNA-α signaling, effectively bypassing the therapeutic block and allowing inflammation to persist. While the specific alternative pathways upregulated in JAK inhibitor non-responders remain unexplored, studies investigating anti-TNF therapy resistance have shown an expansion of IL-23 receptor-expressing T cells in anti-TNF refractory Crohn’s disease patients [91]. This finding highlights IL-23 as a potential therapeutic target for these patients. If TNF-α signaling is found to be upregulated in JAK inhibitor-refractory patients, combining JAK inhibitor therapy with anti-TNF treatment could potentially enhance therapeutic efficacy. However, no studies have directly compared this combination to JAK inhibitor monotherapy. Although limited clinical evidence suggests that the combination is safe, concerns regarding increased infection risk persist [92]. JAK inhibitors may also be combined with other biologics such as vedolizumab (anti-α4β7-integrin) or ustekinumab (anti-IL-12/23) to improve efficacy, but evidence is limited and safety concerns remain [92]. Alternatively, JAK inhibitors may be combined with non-specific immune suppressants such as 5-ASAs. A retrospective study in a Japanese cohort found that, in patients using 5 mg tofacitinib twice daily, concomitant use of 5-ASA reduced the relapse risk, but this effect was not observed with 10 mg tofacitinib [93].

Other factors that may influence the efficacy of JAK inhibitors are genetic polymorphisms in genes encoding JAKs, STATs, or other key signaling molecules, as these variations can impact drug targets or downstream signaling. Epigenetic modifications, which alter gene regulation without changing the underlying DNA sequence, could further affect a patient’s sensitivity to JAK inhibitors. Additionally, the gut microbiota plays a crucial role; its composition and activity can influence both local inflammation and the metabolism of drugs, potentially modulating the effectiveness of JAK inhibitors. Nutritional status and diet may also be significant, as nutrient availability can impact immune responses and drug metabolism, adding another layer of complexity to treatment outcomes [94]. In summary, resistance to JAK inhibitor treatment may be driven by baseline JAK/STAT pathway signaling in immune and stromal cells, expression of transporters of JAK inhibitors, activation of alternative inflammatory pathways, and other relatively unexplored factors.

## 6. Conclusions and Future Perspectives

JAK inhibitors decrease the abundance of immune cells including neutrophils, macrophages, monocytes, dendritic cells, Tregs, plasma cells, and dendritic cells, as well as inflammatory fibroblasts in the colons of IBD patients who respond to this treatment. These therapies also increase epithelial cell abundance and improve intestinal barrier integrity. However, these inhibitors impair wound healing and regeneration in mouse models, although the relevance of these findings to humans remains uncertain due to biological differences in regenerative mechanisms. Additionally, JAK inhibitors suppress key inflammatory functions of epithelial cells in UC, with emerging evidence pointing to specific effects of JAK1 inhibitors.

Despite these advances, the precise roles of different cell types in resistance to JAK inhibitor therapy are not yet fully understood. Baseline JAK/STAT pathway signaling in immune and stromal cells and expression transporters of JAK inhibitors appear to influence therapeutic response. Yet, the multifaceted nature of UC pathogenesis and treatment resistance suggests that additional factors—such as compensatory pathway activation, genetic polymorphisms, epigenetic modifications, and the gut microbiota—also play significant roles.

Given the heterogeneity of UC and the complex interplay of contributing factors, future research should adopt comprehensive omics approaches to better capture the nuances of disease biology. For instance, Mo et al. utilized transcriptomics to identify three molecular subtypes of UC with heterogeneous immune profiles and distinct response rates to biologic therapies [8]. This kind of molecular stratification could help to personalize treatment options and improve response rates. Furthermore, spatial analysis techniques —such as those used by van Unen et al. to map disease-associated immune cell networks in IBD—could shed light on the local tissue microenvironment and the interactions between cell populations [95]. These insights might uncover the mechanisms driving resistance and suggest novel therapeutic targets.

## Data Availability

No new data was created or analyzed in this study. Data sharing is not applicable to this article.

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
