# Peer review of "The Cell-Specific Effects of JAK1 Inhibitors in Ulcerative Colitis"

_jcm, 2025, doi:10.3390/jcm14020608_

Round 1
Reviewer 1 Report
Comments and Suggestions for Authors
This is an interesting review addressing the role of immune, epithelial, and stromal cells in response to JAK1 inhibition with potential strategies to improve treatment by a tailored approach, such as predicting responders and identifying complementary therapeutic targets. They discussed current treatments with JAK inhibitors, such as tofacitinib, filgotinib and upadacitinib and their differences as well as treatment response and failure.
The manuscript is well organized and the topic of clinical interest. However, in my opinion, it can be improved by expanding the section addressing the crucial role of Janus kinase inhibition on barrier and mucosal permeability and integrity. Intestinal barrier integrity is a critical factor in immune-mediated chronic intestinal disorders such as inflammatory bowel diseases (IBD) as well as celiac disease (CD). In particular, it has been previously demonstrated that tight junction function and regulation alteration is an early step in the pathogenesis of the most common immune-mediated inflammatory diseases (IBD and CD).
The increased intestinal permeability may trigger the immune response against microbiota antigens that, in turn, sustains and enhances inflammatory pathways contributing to the mucosal damage. Importantly, as a consequence of the increased mucosal permeability and immune response promotion, serum antibodies against microbial antigens such as Anti-Saccharomyces cerevisiae (ASCA) appear and their serum titers are related to disease activity as previously demonstrated in IBD and celiac disease (Aliment Pharmacol Ther. 2005;21(7):881-7. doi: 10.1111/j.1365-2036.2005.02417.x.). Importantly, the appearance of ASCA, reflecting the increased mucosal permeability, preceed the clinical onset of disease as reported in celiac disease patients (Anti-saccharomyces cerevisiae antibodies (ASCA) in coeliac disease. Gut. 2006 Feb;55(2):296.). In celiac disease patients, serum ASCA disappear afetr gluten-free diet initiation as a results of mucosal recovery.
The clinical application of serum ASCA in the management of IBD and celiac disease patients should be recalled to increase the clinical impact of the manuscript.
This important role of Janus kinase inhibition on barrier and mucosal permeability has currently under investigation as potential target of novel treatment for celiac disease, as described (J Clin Immunol. 2013;33:586-94. doi: 10.1007/s10875-012-9849-y; Drug Discov Today. 2024 Sep;29(9):104113. doi: 10.1016/j.drudis.2024.104113.).
Recently, the critical roles of janus kinase/signal transducer and activator of transcription signaling pathways, involved in the secretion of more than 50 cytokines, has been extensively discussed not only in autoimmune diseases but also in tumorigenesis and thus might have an important role of tumor-related complications of IBD and celiac disease (Front Pharmacol. 2024 Jan 3;14:1326281. doi: 10.3389/fphar.2023.1326281.)
Reviewer 2 Report
Comments and Suggestions for Authors
Dear Editor
This is a well-written paper regarding the role of JAK in UC. The following are my comments.
#1. Line 417, you mentioned alternative pathways might compensate for JAK inhibition. Could you further explain alternative pathways in more detail or provide a figure for this point?
#2. Have you reviewed studies regarding JAK inhibition to reduce the development of colitis cancer?
#3. Do you consider a synergistic effect of JAK inhibition and the co-therapy of 5-ASA?
#4. As anti-TNF, anti-integrin, and anti-IL12/23 agents were now widely used for UC therapy, the article only reviewed the effect of anti-TNF and subseqquent JAK therapy. How about other the effect of other biologics ?such as anti-integrin failure and their response to JAK ?
Round 2
Reviewer 1 Report
Comments and Suggestions for Authors
The Authors satisfactorily addressed the raised issues.
Reviewer 2 Report
Comments and Suggestions for Authors
Dear Editor
The authors response to questions raised well. I have no more comments.